# Understanding barriers to implementing referral procedures in the rural and semi-urban district hospitals in Tanzania: Experiences of healthcare providers working in maternity units

Lilian Mselle[1]*, Nathanael Sirili[2], Amani Anaeli[2], Siriel Massawe[3]

1 Department of Clinical Nursing, School of Nursing, Muhimbili University of Health and Allied Sciences, Dar es Salaam, Tanzania, 2 Department of Development Studies, School of Public Health, Muhimbili University of Health and Allied Sciences, Dar es Salaam, Tanzania, 3 Department of Obstetrics and Gynaecology, School of Medicine, Muhimbili University of Health and Allied Sciences, Dar es Salaam, Tanzania

* nakutz@yahoo.com

## Abstract

### Introduction

Maternal and perinatal deaths occurring in low and middle income countries could be prevented with timely access to maternal and new-born care. In order to increase access to maternal and child health services, a well-functioning referral system that allows for continuity of care across different tiers of healthcare is required. A reliable healthcare system, with adequate numbers of skilled staff, resources and mechanisms, is critical to ensuring that access to care is available when the need arises.

### Material and methods

This descriptive, qualitative study design was used to explore barriers to implementing a reliable referral system. Twelve individual qualitative interviews were conducted with health care providers working in rural and semi-urban district hospitals in the Northern, Western, Eastern and Southern zones of Tanzania. Thematic analysis guided analysis of data.

### Results

Three (3) main and interconnected themes were abstracted from the data relating to participants' experiences of referring women with obstetric complications to adequate obstetric care. These were: 1. Adhering to a rigid referral protocol; 2. Completing the referral of women to an adequate health facility and 3. Communicating the condition of the woman with obstetric complications between the referring and receiving facilities.

### Conclusion

Because of referral regulations, assistant medical officers were unable to make referral decisions even when they felt that a referral was needed. The lack of availability of hospital

**Data Availability Statement:** All relevant data are in the paper and its supporting information files.

**Funding:** Authors are very grateful to the Translating Research into Action (TRACTION) Project in the United States, for initiating the study as part of a larger inter-country study in Malawi and Tanzania, and for financial support. However, the project did not play any role in the study design, data collection and analysis, decision to publish, or preparation of the manuscript.

**Competing interests:** The authors have declared that no competing interests exist.

transport as well as the lack of a reliable feedback mechanism, prohibited effective referrals of patients. The Ministry of Health should revise the referral protocol to allow all clinicians to provide referrals, including assistant medical officers- who make up the majority of clinical staff in rural health care facilities. A mechanism to ensure effective communication between the referral facility and the tertiary care hospital should be instituted for quality and continuity of care. Furthermore, health care facilities should put aside budget for fuelling the ambulance for effective referrals.

## Introduction

Nearly all maternal and perinatal deaths occur in low and middle income countries [1]. The underlying causes of this large mortality burden in the intrapartum period include; obstetric haemorrhage, hypertensive disorders, obstructed labour and infections which directly impact maternal health. These complications also lead to hypoxic-ischaemic insults which either directly cause perinatal deaths [2] or lead to long-term disabilities in the infant [3]. These medical complications leading to increased maternal and perinatal morbidity and mortality could be prevented with adequately trained staff, sufficient resources and successful referral mechanisms to ensure prompt access to emergency obstetric care by women when needed.

Access to emergency obstetric services is integral to a fundamental human right [4]: the right to achieve the highest attainable standard of health. Central to the attainment of this right within a hierarchical health system is a well-functioning referral system that allows for continuity of care across different tiers of care. The referral system in Tanzania is organized into a pyramid; starting from the health post, dispensary and health centres that provide Basic Emergency Obstetric Care (BEmOC) and treatment of minor conditions. At the district level, there are the district hospitals where drugs, equipment and skilled staff are available to provide Comprehensive Emergency Obstetric Care (CEmOC). There are also regional hospitals in each region, with the highest levels being national and specialized hospitals [5, 6].

Despite ongoing improvement in healthcare facilities including the upgrading of health centers in order to provide comprehensive obstetric services, maternal mortality has remained unacceptably high. By 2015, the maternal mortality rate in Tanzania was estimated to be 556 maternal deaths per 100,000 live births [7]. Among the factors contributing to Tanzania's high maternal mortality rate is a weak referral system [8]. An efficient and effective referral system is critical to respond to complications which necessitate the transfer a woman and/or newborn with intrapartum complications to the next level health facility.

The three delays model suggested by Thaddeus and Maine identifies barriers in the different phases of birth care seeking that can delay access to effective interventions to prevent maternal mortality [9]. These include; (1) delay in recognizing danger signs and deciding to seek care, (2) delay in reaching healthcare facility for appropriate care and (3) delay in receiving appropriate care once a facility is reached. To facilitate an effective referral system, the Ministry of Health, Community Development, Gender, Elderly and Children (MOHCDGEC) prepared a health care referral guideline that adheres to the procedural components of an efficient and effective referral network that include: 1. Defining the basis for referral, 2. Improving the documentation on the referral form, 3. The mode and speed of transportation between facilities, and 4. The communication and feedback to the referral facility [10]. Improvement of the referral system would contribute to two of the United Nation's Sustainable Development Goals: Goal 3; ensuring healthy lives and promoting wellbeing and Goal 9; building resilient infrastructure [11] and realization of full continuum of obstetric care [12].

The focus of studies conducted on the referral system for maternal and newborn services in Tanzania have been on issues related to decision making and the acceptance of referral advice [13, 14], effectiveness of the maternal referral system in a rural setting [15], their referral patterns [16] and the reasons women were not adhering to referral advice [17]. Another study looked at delays of referring women with higher levels obstetric complications has been reported to be associated with barriers of referral resulting from challenges of transportation [18]. Although common barriers to successful referral are generally known, the relative importance of these barriers in Tanzania should be identified in order to guide the design of appropriate interventions to improve referrals. This study describes barriers to the implementation of a referral system as described by experienced health care providers in rural and semi-urban district hospitals in Tanzania.

## Material and methods

### Design and participants

This study was part of a large study which adopted a case study design to explore barriers and facilitators for performance of Caesarean section by Assistant Medical Officers in Tanzania. The descriptive phenomenology study design [19] was carried out in four districts in the Northern, Western, Eastern and Southern zones of Tanzania. The zones were purposely selected because they had caesarean section rates either above or below the national average. Further, the type of health facility (run by government or a faith-based organisation), and the location of the facility (rural or urban) were considered in order to tease out the factors that determined whether the referral protocols and system facilitated women's access to comprehensive emergency obstetric services. Specifically, this study was conducted in 6 health facilities; including both Government and Faith-based organisations, and categorised as either rural or semi-urban (see Table 1). Although the healthcare facilities involved in the study were located in rural or semi-urban areas, they provided either basic emergency obstetric and newborn care (BEmONC) or comprehensive emergency obstetric and new-born care (CEmONC). However, due to various reasons they are compelled to refer patients to other healthcare facilities for appropriate obstetric care. The common reasons for referral were; inadequate number of CEmONC trained health care providers, limited (lack) of medical equipment and insufficient supply of essential obstetric medicines such as safe blood transfusion and magnesium sulphate [18, 20]. In Tanzania all healthcare facilities whether public, private or faith-based are implementing Government policies including maternal and neonatal referral guidelines.

The study participants included Medical Officers, Assistant Medical Officers (AMOs), Nurse-Midwives and Health Secretaries working in the 6 healthcare facilities and District Medical Officers (DMOs) of the 4 study districts and were conveniently recruited. Because of the limited number of medical personnel in these study settings, convenience sampling strategy was used and therefore whoever was around during data collection period and willing to

**Table 1. Zones, region and type of healthcare facilities involved in the study.**

| Zone | Region | Type of the facility |
|---|---|---|
| Northern | Tanga | Government- Rural District Hospital |
| Eastern | Morogoro | FBO- Semi-urban |
| | | FBO- Rural Health Centre |
| Southern | Mtwara | Government-Rural District Hospital |
| Western | Kigoma | Government-Rural District hospital |
| | | Upgraded Health Centre |

participate in the study was recruited [21]. Heads of the maternity services assisted with identification of participants based on set inclusion criteria. The participants had to work in the labour ward for 3 or more years to ensure that they had adequate experience managing women with obstetric complications and utilizing the referral system or had administrative roles in the maternity unit. The researchers met the labour ward staff, then provided information about the study purpose and issues of confidentiality. Thereafter, a convenient interview time was arranged with those who agreed to participate in the study.

## Interviews

Twelve (12) individual, qualitative interviews were conducted using an in-depth interview guide. The guide was developed from a review of the literature that explored barriers to the implementation of a referral system, and focused on the experience of healthcare providers in rural and semi-urban district healthcare facilities for women. In order to avoid interference with the activities of the healthcare facilities, interviews were conducted when the participants were off duty. Interviews were conducted in Kiswahili in a quiet and comfortable room within the healthcare facility. At the end of each session, the recorded interviews were listened to allow new emerging issues to be included in the guide before the subsequent interviews [22]. After 10 interviews were conducted, it was noted that the knowledge saturation was reached, yet, 2 more interviews were done to adhere to the recommendations that 12 interviews will be required to achieve saturation [19, 23]. The sample size for qualitative studies depend on the complexity of the research questions, the purpose of the research study, the diversity of the sample, the nature of the analysis and the time and resources available [24–26]. On average, each interview took about 50 minutes. Before the interview began, participants signed the consent form to agree to participate and to allow recording of their conversations during the interview sessions. All participants were informed on the study aim, procedures and that their participation was purely voluntary so they were free to decline or withdraw at any time in the course of the study. The study was approved by the Senate Research and Ethics Committee of the Muhimbili University of Health and Allied Health Sciences (MUHAS) and the Ministry of Health Community Development Gender Elderly and Children (MOHCDGEC).

## Analysis

Analysis of the data was guided by Braun and Clark thematic analysis, a flexible data analysis method that focuses in identifying themes and patterns [27, 28]. The analysis followed the six-stage in thematic analysis [20]. First, the recorded interviews were listened twice and thereafter transcribed verbatim and translated into English. The translated transcripts were then cross-checked against the Kiswahili transcripts to ensure the accuracy and completeness of translations before coding. When discrepancies occurred, these were corrected accordingly. Second, initial codes were extracted from the transcripts, whereby each researcher coded one transcript. After the team agreed on the codes, the NVivo10 software was used to organise the codes and categories. As the process continued, new, emerging codes were formulated and compared with the initial codes. Third, involves sorting the codes based on their similarities and differences into potential categories and themes, and collating all the relevant coded data extracts within the identified themes. Fourth, themes were reviewed by either splitting, combining, or discarded them during the process. Fifth, the themes and categories were named and were discussed and agreed upon among the team. Final stage, the report was produced in which a few quotes were included to demonstrate the essence of a point.

## Results

### Participant characteristics

The 12 health care providers interviewed were between 34–59 years of age, 6 were medical practitioners (MD or AMO), 5 were nurse-midwives working in the labour ward and 1 was a health secretary. Seven 7 providers had worked in the wards where they had interviewed for 7 years or more.

### Themes

Three (3) main and interconnected themes and subthemes were abstracted from the data relating to participants experiences referring women with obstetric complications for obstetric care: 1. Adhering to a rigid referral protocol; 2. Completing the referral of women to an adequate health facility and 3. Communicating the condition of the woman with obstetric complications (see Table 2).

### Adhering to a rigid referral protocol

**Failing to refer women directly to an adequate health facility.**    As per the Tanzania referral procedure, patients from health post, dispensary and health centre are referred directly to the district hospital. District hospitals refer patients to regional hospitals which can refer patients to the national or specialised hospitals (higher level) if need arises. This rigidity in the path of referral was reported as a challenge by the health care providers especially when they know that the required service for such patient is not available at the regional hospital. This system cause unnecessary delay for accessing appropriate services to the referred patient:

*(. . .) the big problem is whenever a woman requires referral for a particular service, at the district hospitals we cannot refer patients directly to the national level. We are required to refer her to the regional hospital even though sometimes the required service is not available at the regional hospital. (. . .). Therefore, the woman will just go there because it is a must for the referrals to pass through the regional hospitals. This system puts obstacles and limitations in accessing services in time* [Medical Officer].

*(. . .) our referral centre is Bugando Hospital, but there are times when you know that this particular service is not available in that hospital but as per the guidelines we have to refer them to the next level referral facility first regardless of whether the service is available or not. Therefore referring a patient to a hospital that has no particular service is unnecessary circle that will delay a patient from receiving services in time.* [Assistant Medical Officer].

**Table 2. Themes and subthemes from healthcare providers experience in implementing referral guidelines.**

| Themes | Sub-themes |
|---|---|
| Adhering to a rigid referral protocol | i. Failing to refer women directly to an adequate health facility |
| | ii. Referral is only made by the medical doctor |
| Completing the referral of women to an adequate health facility | i. Lack of hospital ambulance |
| | ii. Delayed repair or replacement of broken ambulances |
| | iii. Shortage of funds to fuel ambulances |
| Communicating the condition of the woman with obstetric complications | i. Lack of advanced notice about the referral of women to the recipient healthcare facility |
| | ii. Lack of communication feedback about the women's condition |

**Referral is only made by the medical doctor.** According to the referral guidelines, only a medical doctor (MD) is authorised to fill/complete the referral form providing detailed information of the patient diagnosis, management given and the reasons for referral. Participants were aware of the referral regulation which was very frustrating in their desire to provide timely care particularly when MDs were unavailable:

*(. . .) usually an assistant medical officer can't say this patient has to go to the next level health facility without the medical officer's approval. If there is an emergency that has happened during the night and there is a need for referral that very night a medical officer must be notified (. . .). The referral plan will be initiated but I cannot write a referral, the medical officer is informed first and he will have to come to write/sign a referral* [Assistant Medical Officer]

*There are some circumstances that the assistant medical officer may write a referral letter but the medical officer in charge must be around during that time to sign the form, especially for obstetric cases he must be there.* [Nurse-midwife]

It was learned that occasionally the clinical officers (CO) or any health care provider who will be in charge at the referring facility may write a referral letter or fill a referral form especially when an assistant or medical doctor is not present:

*(. . .) Sometimes it happens that the medical officer is not around, which happens so often, and sometimes even assistant medical officers are not present, we call a clinical officer because others are not there, and they come to prepare a referral.* [Nurse-midwife]

*The assistant medical officer in this facility is the one who provides the final approval for the patient to be taken to the higher level health facility as there is no medical officer here. From here we refer patients to a district hospital. His role is to examine the patient and write a referral letter for the patient to go to the regional hospital.* [Assistant Medical Officer]

*Assistant medical officer gives referral just like medical doctors if they assess the patients' condition and may refer her to the next level of service as medical doctors.* [Health Secretary]

Earlier on when the number of medical officers was limited, many health facilities were manned by assistant medical officers, clinical officers or nurses. During that time, the clinical officer in charge of the facility was responsible for referring patients to the next level:

*(. . .) Previously the in-charge was responsible to fill the referral forms. The in charge could be someone else who is not an AMOs, he could be a clinical officer or even a nurse. . . other facilities have nurses only, and the in charge is a nurse and no doctors.* [Medical Officer]

*Previously AMOs were at the top and had the final say on referral, but now they cannot do that, they have to consult or call a medical doctor for approval. I think those are the few changes which have occurred currently. (. . .) now I will not be referring patients anymore.* [Assistant Medical Officer]

## Completing the referral of women to an adequate health facility

**Lack of hospital ambulance.** To complete a referral of patients from a lower level facility to a higher one, a functioning vehicle or ambulance is necessary for prompt access to emergency obstetric care. In this study, we found that the availability of functioning vehicles to transport women with obstetric complications was a major impediment to completing a

referral. The lack of hospital ambulances was reported to be a major barrier in the successful referral of patients to an adequate healthcare facility:

> *The main challenge we face now is that we do not have an ambulance to transport patients to the regional hospital. (. . .) we usually ask the patient and relatives to find an alternative transport.* [Nurse-midwife]

Successful management of obstetric emergencies depends on how promptly the woman is able to get into the health facility for adequate obstetric care. Participants reported that health facilities had ambulances but that most of these were used for administration activities. For example, one participant lamented that the only vehicle available in the facility to shuttle patients was taken by the council for other administrative activities. Because of this, it was not possible to transport women with obstetric complications to an adequate health facility for further management:

> *(. . .) previously each health centre had a vehicle and it was easy to transport patients but later it was taken by the district council for special duties because the District does not have a vehicle. . . There are three health centres in this district council. I do not think there is any health centre that has a vehicle for patients right now.* [Assistant Medical Officer]

Because of lack of transport, many women arrived at higher level healthcare facilities when they were in serious condition:

> *(. . .) some women arrive while they have ruptured the membrane because of long travels, others' live about 7 kilometres away, like those coming from Msagati, their roads are in poor shape especially during rainy season, (. . .).* [Nurse-midwife]

The referral protocol requires that the referral form provide detailed information about the patient and that the referred patient should be accompanied by a health care provider. But on some occasions, the referral protocol was not followed due to the lack of transportation. When this occurred, women were asked to go unaccompanied to a higher level healthcare facility without a companion or a referral note:

> *(. . .) at times we do not follow protocols, when you don't have an ambulance you just ask them to go even without a nurse and sometimes without documentation because when you document you will have to have someone to accompany the patient to the recipient facility. So we just take a brief history and inform the patient that the doctor is currently not available so just go to another facility. I think this is not right.* [Medical Officer]

**Delayed repair or replacement of broken ambulances.**   Participants also reported that some health centers had ambulances, however, as time went by they broke down without repair or timely replacement:

> *The main challenge is transportation, when the ambulance breaks it can take long time to repair, it may take two to three months to get repaired.* [Medical Officer]

> *We had an ambulance after it broke down two years ago and taken to the district for repair it was never returned. So when we need to refer a woman we ask for help from the district*

*hospital. Sometimes we get and sometimes we do not, in which case we ask the patient and relatives to look for other means to get to the hospital.* [Assistant Medical Officer]

*(. . .) the facilities are not enough, today the ambulance maybe working, on the next day it may not, so that is how it is and a nurse has to escort the patient, so when you look at it, it is real a challenge unless the patient pays transport fee for the nurse as well to accompany her.* [Medical Officer]

The repair of the government vehicle is usually done through an agent. But it is common that payments for repair are delayed because the funds designated to pay the agent have been used to purchase more pressing matters such as medical supplies, consequently the payment to repair broken vehicles is not done on time:

*(. . .) we face problem when the vehicle is taken for maintenance and the funds that was meant to pay for repair has been used for other things. So an agent fails to repair it because he is not paid.* [Health Secretary]

**Shortage of funds to fuel ambulances.** Ambulances may be available in the healthcare facilities; however, they may not be running because of lack of fuel. In such situations, the health care providers have no option than asking patients to pay for the cost of fuel to be transported to the next higher level facility for adequate management:

*(. . .) there are few challenges regarding referral for emergency obstetric care. . . you may find out the ambulance has no fuel so we have to ask patients to contribute for the fuel cost in order to be taken to the hospital.* [Assistant Medical Officer]

*(. . .) the main issue is fuel, because when you refer the patient the vehicle must have fuel so that you can refer the patient. When there is no fuel it is impossible, unless the patient can afford to purchase fuel. For example, if the patient is referred to the regional hospital here, the distance to get there and return you need about 80 litres. Therefore, out of 80 litres the hospital can provide 40 litres and relative will be asked to contribute another 40 litres.* [Nurse-midwife]

*(. . .) we offer her a car and a driver the remaining things are upon the patient, she will have to purchase the fuel (. . .) she will have to provide twenty litres.* [Health secretary]

On many occasions, women were unable to pay the cost of fuel. When this occurs, the healthcare providers are forced to use their out of pocket money to buy fuel for the women to be transferred:

*(. . .) some of the relative have no cash so they would tell you to do anything because they don't have cash even for their upkeep and the patients can't go alone so those are the challenges we are facing.* [Medical Officer]

*The fuel to run the vehicle is the major obstacle. You may have a patient in need and whom you are sure that they will benefit more if they are referred to the regional or national hospital, but you find out that the family is not capable of handling the costs (. . .). Often we use our own money to transport patients.* [Assistant Medical Officer]

Finally, when a hospital vehicle is unavailable because of either lack of fuel or need for repair, women may be forced to use public transportation despite their clinical condition:

*(. . .) There are no ambulances in peripheral facilities [i.e. dispensaries and health centers] so patients are brought here by either public transportation, motor cycle or a Bajaji, which is uncomfortable and cause a lot of delays as patients arrive here in worse condition and some with complications already, giving us a hard time in their managing.* [Assistant Medical Officer]

*Patients get hard time here; there are no taxis so they have to use Bajaj together with the escorting nurse. You can imagine the trouble they get, sometimes an escorting nurse has to take a motorcycle (. . .).* [Medical Officer]

## Communicating the condition of the woman with obstetric complications

**Lack of advance notice about the women being referred to the recipient healthcare facility.** Information on the patient's condition before transfer to the adequate healthcare facility is of paramount importance. Health care providers awaiting the transfer in the recipient healthcare facility may then prepare themselves and other necessary staff before the patient arrives. The participants in this study reported the importance of communication with health care providers in the recipient health care facility before the patient is transferred. However, this was an uncommon practice because the health facilities often lacked a reliable means of communication:

*Another challenge is telephone communication; ideally we are supposed to inform providers in the regional hospital before a patient is moved there. We have to call and inform them that a mother with certain symptoms is referred there, so that they could prepare and be ready to receive the patient.* [Assistant Medical Officer]

**Lack of communication feedback about the women's condition.** Participants were also aware of the importance of referral feedback. There is a special feedback form where the doctor caring for the woman in the recipient healthcare facility documents her clinical care and subsequent condition and shares it with the providers who initially referred the patient. This feedback loop is critical to ensure continuity of care. However, the feedback forms were often not used:

*"(. . .) there are special forms for feedback but there are little possibilities of getting feedback, there are special form written feedback from referral health facility, but it is very difficult for a person to return to you. I have not received any feedback from the district hospital, but I think it is a good thing (. . .).* [Assistant Medical Officer]

*I have never seen it, once we refer we are done with that business unless it is of a patient you have special interest where you can call a friend there and ask the ABC of that patient.* [Nurse-midwife]

*Mmh you mean feedback about the patient who has been referred to the hospital? (Yes) honestly, there is no feedback; after the patient is sent there, that's it. We have no system of receiving feedback or follow up, once we send them that's it.* [Health Secretary]

The doctors from the referring health facility only get information about whether or not the patient arrived, and about what transpired during the transfer from the escort nurse when he or she returns. Moreover, when there is feedback, it is occasionally given as a complaint when the patient's condition deteriorates at the higher referral facility:

*The woman usually is accompanied by a nurse with the referral form. After getting to the recipient health facility the escorted nurse will be informed if the woman is accepted or not (. . .). There is no any other information given to the referring doctor, the only information offered is that by the nurse who escorted the woman.* [Nurse-midwife]

*We will only get feedback if the patient was mismanaged and let's say the patient died that is when we will get feedback and the feedback is usually condemning that we are useless and completely irresponsible, you see. But, if the management was successful nobody will inform you.* [Assistant Medical Officer]

*Honestly there is no feedback maybe if the doctor is so concerned about the patient he can call to ask for the wellbeing of the patient or the recipient hospital may call to give feedback when death occurs for the sake of auditing.* [Medical Officer]

It was further learned from this study that, in the referral regulations, there is no guidance about how best to communicate the management of the patient to the referring doctor. There is lack of a clear feedback mechanism as well as instructions on how to go about it including feedback forms.

*(. . .) there are special forms for feedback but rarely do we send them back. (. . .) it is not clear who returns the completed feedback form back to the healthcare facility that referred the patient. So the problem is how do we return it, should you ask the patient to return it or how would you return it, it is a problem.* [Assistant Medical Officer]

*(. . .) that is the area which has small weakness, when they discharge her [patient], you should write to the person who referred that patient letting him know that the patient was received and was managed very well. We did, 1, 2, 3, and now we are sending her back to him [doctor] for continued care. Sadly we do not have any feedback arrangements, (. . .).* [Medical Officer]

## Discussion

### Referral and the referral protocol

Consistent with other studies [29, 30], the findings from this study indicate that the implementation of the referral guidelines is not followed. For example, the national referral regulations require that medical doctors sign the referral form in order for the patient to be transferred to the next level health facility for adequate or speciality care. However, because of the limited number of medical doctors in health facilities in the districts, many facilities are manned by the Assistant Medical Officers—a physician assistant-like professional who is an integral part of community healthcare [31]. Therefore, it is impractical to authorise only medical doctors to refer patients, in case of emergency other medical personnel should be allowed to sign referrals to save lives. Lack of specialists was reported as a major reason for lack of referral implementation in Iran [29]. The reason behind mandating only medical doctors to refer patients was to ensure that the transferred patient actually required specialised care, since medical doctors are believed to highly be clinical competent than other health care providers.

While the MOHCDGEC referral regulation does not authorize the AMOs who are the majority in health care facilities in the districts to sign referral forms, it was reassuring to learn that in the absence of medical doctors the AMO, clinical officer or a nurse could sign a referral form for the patient to be transferred to the next level health care facility. In a situation where there is only one medical doctor, then the guideline needs to be followed, often resulting in a delay in the transfer of patients who need urgent care. In the event of an obstetric referral,

there is either a prediction that a woman may go on to develop obstetric complications or the complication has already developed and cannot be managed at a specific health care facility [32]. The decision to make a referral is dependent on; the clinical skills of the referring health care provider, the tools for diagnosis, and the availability of a health facility with the necessary specialty care. Other factors that need to be considered are the quality of care at the referral facility, the cost of care, the distance between facilities, the availability of reliable transportation, the availability of clear communication between providers, someone to accompany the patient as well as the feasibility of travel by the patient [33]. The right to the highest attainable standard of health is a fundamental human right [4]. In hierarchical health systems such as in Tanzania, a well-functioning referral system that allows for the continuity of care across different tiers of care is central to achieving this right [34].

Another barrier to safe and efficient referrals for complex patients is that; the national referral protocol requires the doctor to only refer patients to the next level referral health facility even when it does not have the required services. The referral system is designed to optimize the use of three levels of health services and to avoid unnecessary congestion and waste of human and material resources in the specialised levels [35]. The referring clinician should be given a leeway to refer patients to an adequate health facility if the next immediate level health facility has no services that would benefit the patient. This will ensure that women get appropriate management in a timely manner, contrary to rigid referral guidelines that mandate a linear referral procedure that does not meet the emergent clinical needs of the patient. This practice could be the reason for patients not following the referral system, as noted by Eskandari and colleagues [29].

## Transport

The successful management of obstetric emergencies depends on how promptly the woman is able to get into a health facility for adequate obstetric care. Consistent with the findings from studies in Tanzania [18, 36] and elsewhere [37, 38], our study found that moving patients from one facility to another was a challenge. Frequently, women were not able to be moved to the next level health facility due to unavailability of the hospital vehicle or lack of fuel to run the ambulance. Patients were either compelled to travel using unsafe public transportation or buy fuel needed to run the hospital ambulance (if one was available), or at times the health care providers were compelled to use their own limited funds to transport patients to the next level facility. This commonly resulted in some practitioners being hesitant to refer women for adequate care leading to delays occurring in the healthcare facility [9]. Referral regulations also require the woman to be accompanied by a health care provider that makes unavailability of hospital ambulance more critical. The lack of reliable transportation to shuttle women to higher level healthcare facilities in the event of an obstetric emergency may contribute to the increased number of maternal deaths in these districts. The WHO describes the consequences of inadequate transportation for the delivery of health care to the most impoverished settings causing delays in both seeking and receiving care at health facilities. Also, it is a contributing factor to subsequent deaths and disabilities among women with obstetric complications during labour. The WHO estimated that 75% of maternal deaths can be prevented through timely access to childbirth-related care. Therefore, the transportation plays an important role in achieving sustainable development goal no 3 of reducing the global maternal mortality rate to less than 70 per 100,000 live births by 2030 [11]. For instance, India has significantly increased institutional deliveries and reduced maternal deaths through implementation of emergency transportation system a model that could be adopted to improve healthcare facilities referral [39].

## Communication during referral

The communication of a patients' clinical information at the time of a referral is essential to the delivery of high-quality health care. Health care providers in primary, secondary and tertiary care facilities value this information exchange for shared patients [40]. Our study found gaps in referral information on patients condition before transfer to the recipient hospital, and lack of feedback to the referring hospital following treatment. Health care facilities lack reliable communication mechanism to facilitate a swift transfer into a referral health care facility, as well as a mechanism to provide information on the patient's management following referral consistent with studies reported elsewhere [29]. Some studies have proposed the use of software, phones, emails and texts for sharing patient information while maintaining confidentiality [41]. However, these tools are usually not available in most rural health facilities and some do not positively impact referral communication [42]. The number of referral patients, the limited number of medical practitioners and the cost of the technologies are likely to make these possible solutions impractical. However, having the appropriate clinical information about a woman's condition prior to her transfer would help staff in the recipient facility prepare and enhance the quality of care. Evidence show that specialists who received timely patient referral information prior to referral were able to provide optimal care twice as often as specialists who did not [40].

Although the MOHCDGEC referral regulation requires feedback to the referring health facility after the management of the patient is complete, it does not specify nor suggest a mechanism to do so. In practice, the patient is usually discharged home with a discharge summary that provides little detail of what was done. In addition, many hospitals use health management systems that are not linked across health care facilities. All of this results in a lack of feedback to the referral health facility, consequently poor continuity of care. The referral process is a critical component of quality care and this breakdown in communication needs to be improved in order to achieve better maternal and newborn outcomes [43].

## Methodological considerations

The rigor of data was increased through credibility, dependability, conformability and transferability [44]. Validation of the key themes through dialogue with all members of the research team, as well as continuous reflection and revisiting codes to ensure accurate fit [45]. For dependability, all transcripts were retained for review and served as an audit trail. In addition the researchers were receptive to the participants' ideas to reduce bias in the findings. To ensure authenticity and credibility, the study used multiple data sources (i.e., Nurse-midwives, Assistant Medical Officers, Medical Officers and Health secretaries), drawn from diverse regions of Tanzania (i.e., Morogoro, Tanga, Kigoma and Mtwara), from a variety of settings (i.e., rural and semi-urban) and within a wide range of health facilities (i.e., public and faith-based). Nevertheless, there are some limitations to this current study. The analysis of the interviews was completed in English from translated transcripts; the quality of the participants' accounts may be affected, as some Kiswahili words may not have a direct translation in English. However, the transcripts were verified by the research team fluent in Kiswahili and then back translated from English to Kiswahili to check the quality of translation and ensure that the translations were accurate. Further, all codes and themes were discussed among the researchers who were able to review the original transcripts. The direct quotes provide in-depth and rich data and could be used as transferable knowledge to similar contexts to that of Tanzania. Although in-depth interviews with a small sample size of healthcare professionals can be considered as strength, we also recognize it is a limitation as we involved only six healthcare facilities located in four different regions, and that we did not include the

experiences of healthcare providers in the receiving referral healthcare facilities. Therefore, we recommend further studies be conducted in facilities receiving referrals to learn their perspectives and experiences of implementing referral guidelines.

## Conclusion

Because the national referral guideline dictate that only medical officers should write and sign referral forms, the AMOs are unable to make referral decisions even when they trust that a referral is needed. The lack of reliable transportation (absence of ambulances and lack of fuel to run the vehicle), and the lack of a feedback mechanism between hospitals impeded effective referral decisions and continuity of care. The Ministry of Health should revise the referral protocol to include other clinicians (including assistant medical officers who are the majority in rural health facilities in Tanzania) to provide referrals to higher levels of care. The Ministry of Health beside strengthening and upgrading CEmONC facilities, need also to focus on developing a functional and effective transportation system to ensure patients get to the receiving health facilities timely. Further, mechanisms to ensure effective communication from the referring hospital to the higher level of care and feedback following the management of the referred patient back to the original hospital, should be instituted for quality and continuity of care.

## Supporting information

**S1 Data.**
(ZIP)

## Acknowledgments

The authors would like to thank and acknowledge the contribution of the clinicians including nurse-midwives, assistant medical officers, medical officers and health secretaries who gave their time to be interviewed for this study. Further, we are grateful to the Ministry of Health (MOHCDGEC) for providing permission to collect data from the 4 District hospitals. We are very grateful to the Translating Research into Action (TRACTION) Project in the United States, for initiating the study as part of a larger inter-country study in Malawi and Tanzania. Finally, we would like to thank Susan Wood RN, MSN, MPH, IBCLC, a former US Fulbright Scholar teaching at MUHAS for editing the final manuscript.

## Author Contributions

**Conceptualization:** Lilian Mselle, Nathanael Sirili, Amani Anaeli, Siriel Massawe.

**Data curation:** Lilian Mselle, Nathanael Sirili, Amani Anaeli, Siriel Massawe.

**Formal analysis:** Lilian Mselle, Nathanael Sirili, Amani Anaeli, Siriel Massawe.

**Methodology:** Lilian Mselle, Nathanael Sirili, Amani Anaeli, Siriel Massawe.

**Supervision:** Siriel Massawe.

**Writing – original draft:** Lilian Mselle.

**Writing – review & editing:** Nathanael Sirili, Amani Anaeli, Siriel Massawe.

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
