## [Decision Letter · Decision Letter 0]

22 Mar 2021

PONE-D-20-36193

Understanding barriers to implementing referral procedures in the rural and semi-urban district hospitals in Tanzania: Experiences from healthcare providers working in maternity units

PLOS ONE

Dear Dr. Mselle,

Thank you for submitting your manuscript to PLOS ONE. After careful consideration, we feel that it has merit but does not fully meet PLOS ONE’s publication criteria as it currently stands. Therefore, we invite you to submit a revised version of the manuscript that addresses the points raised during the review process. More specifically

1. Please re-align you literature review to some of the international literature to clearly bring out known barriers to referral process and how your work adds to the know body of literature.

2. The methods section needs to be strengthened as per the comments from the reviewers.

3. Would be useful to tabulate your findings according to the situations (types of facilities and providers)- rather than generalizing them for all situations.

4. Link the discussion to findings and provide possible solutions with supporting literature wherever possible. 

5.  the paper needs very good editing and reformatting of references.

We look forward to receiving your revised manuscript.

Kind regards,

Charu C Garg, Ph.D.

Academic Editor

PLOS ONE

Journal Requirements:

2. Please include additional information regarding the questionnaire used in the study and ensure that you have provided sufficient details that others could replicate the analyses. For instance, if you developed a questionnaire as part of this study and it is not under a copyright more restrictive than CC-BY, please include a copy, in both the original language and English, as Supporting Information, or include a citation if it has been published previously.

3. In the Methods, please discuss whether and how the questionnaire was validated and/or pre-tested. If these did not occur, please provide the rationale for not doing so.

4. We suggest you thoroughly copyedit your manuscript for language usage, spelling, and grammar. If you do not know anyone who can help you do this, you may wish to consider employing a professional scientific editing service.  

5a) If there are ethical or legal restrictions on sharing a de-identified data set, please explain them in detail (e.g., data contain potentially identifying or sensitive patient information) and who has imposed them (e.g., an ethics committee). Please also provide contact information for a data access committee, ethics committee, or other institutional body to which data requests may be sent.

5b) If there are no restrictions, please upload the minimal anonymized data set necessary to replicate your study findings as either Supporting Information files or to a stable, public repository and provide us with the relevant URLs, DOIs, or accession numbers. Please see http://www.bmj.com/content/340/bmj.c181.long for guidelines on how to de-identify and prepare clinical data for publication. For a list of acceptable repositories, please see http://journals.plos.org/plosone/s/data-availability#loc-recommended-repositories.

Please follow the PLOS One guidelines at PLOS ONE: accelerating the publication of peer-reviewed science 

Reviewers' comments:

Reviewer's Responses to Questions

**Comments to the Author**

1. Is the manuscript technically sound, and do the data support the conclusions?

Reviewer #1: Yes

Reviewer #2: Partly

Reviewer #3: Partly

2. Has the statistical analysis been performed appropriately and rigorously? 

Reviewer #1: Yes

Reviewer #2: N/A

Reviewer #3: I Don't Know

3. Have the authors made all data underlying the findings in their manuscript fully available?

Reviewer #1: Yes

Reviewer #2: Yes

Reviewer #3: Yes

4. Is the manuscript presented in an intelligible fashion and written in standard English?

Reviewer #1: Yes

Reviewer #2: Yes

Reviewer #3: No

5. Review Comments to the Author

Reviewer #1: I reviewed a manuscript titled " Understanding barriers to implementing referral procedures in the rural and semi-urban district hospitals in Tanzania: Experiences from healthcare providers working in maternity units.

The manuscript is well written

The authors need to correct a few grammatical errors in the manuscript (e.g line 148, 149)

Methodology: The type of sampling strategy used was not clear. The authors stated the definition of convenience sampling on line 129, but did not clearly indicate if this was the strategy used in this study.

References: Please remove the period (.) in the beginning of reference 4,5,9 and 10

Reviewer #2: The study is relevant in the country context and provides policy recommendations to address high MMR in Tanzania and other countries with similar kind of health system challenges. However, the language of the article could have been more reader friendly. I strongly suggest authors to work with a writing editor to address the grammatical errors, improve the flow and readability of the text.

The authors may consider providing more information regarding “where” mothers are dying in the study area, means whether they are dying at home at the facilities, or in transit. Generally, the data related to place of maternal deaths are available in the MDSR (Maternal Death Surveillance and Response) reports. If Tanzania has MDSR system in place, the data could be analyzed to find out the weakest link in the referral system and findings would complement the qualitative findings of the study making it more statistically relevant.

Authors have tried to refer and divide their findings as per the standard “Three Delays" model, however, it could have been more explicit in a simple and standard language.

Authors may please go through an article from Haiti with similar findings, link is given below:

Barnes-Josiah D, Myntti C, Augustin A. The "three delays" as a framework for examining maternal mortality in Haiti. Soc Sci Med. 1998 Apr;46(8):981-93. doi: 10.1016/s0277-9536(97)10018-1. PMID: 9579750.

Reviewer #3: This is an important issue to be raised, timely referrals are important to save lives. this research is based on a very sample sample size (only 12 interviews) and the sample is also opportunistic. would suggest the authors to have a bigger sample to be able to draw conclusions to bring a policy change on referrals.

the authors should do an English language check.

6. PLOS authors have the option to publish the peer review history of their article (what does this mean?). If published, this will include your full peer review and any attached files.

Reviewer #1: No

Reviewer #2: **Yes: **Amrita Kansal

Reviewer #3: No

---

## [Author Response · Author response to Decision Letter 0]

3 Jun 2021

Responses to reviewers’ comments: 

Understanding barriers to implementing referral procedures in the rural and semi-urban district hospitals in Tanzania: Experiences of healthcare providers working in maternity units

Comments from Academic editor – Charu C Garg

Comments Responses Page

Background: 

Lines 92- talks about 2 delays – but mentions only two in the bracket. Make it more explicit 

The description of the 3 delays model has been revised to include the 3 delays, and mentioned that 2 delays (i.e. 2 and 3) are the case in this context 6

Line 95 – state the key point of the MoH guideline that the audience must know with reference to the referral procedure that you talk about changing in the conclusion 

Put you research in perspective, what you know in the field and what you would be studying additional through your research. Some studies are mentioned from Tanzania, may need to elaborate a little more about their findings about barrier and also what do you know from international literature. The MOHCDGEC guideline on the referral protocol and studies on referral conducted in Tanzania have been reflected as suggested 6-7

Material and methods

The facilities chosen have bias towards rural district hospitals. Normally the types of facilities and where they are based make a difference. Need to clarify by increasing your sample with different types of facilities, or make your study conclusions for specific types of facilities and specific regions. Would also be useful to know the differences in these regions in terms of the health system readiness. 

The study areas were deliberately selected as explained in the manuscript. The differences of the health facilities included in this study have also been explained. 

8

Was there a difference in rural and urban referrals – would help to bring it out. However, you only chose one semi urban facility. It would be useful to increase your sample especially in urban areas to bring out the implementation barriers in urban areas, if that is something that you think will be useful The health facilities included are similar in terms of maternal and child services offered and they all implement referral protocol guideline. This information is included in the manuscript 8

Overall based on interviews of 12 personnel from 4 regions and specific facilities, may not be enough to make general conclusions about the regulatory policy for all types of facilities. The sample size need to be more structured and increased or make the conclusion for characteristics of specific region and facilities used in your study This study uses qualitative approach that primarily focuses on information not individuals. Thus, the sample size is guided by information saturation. Further, the sample size for qualitative studies depend on the complexity of the research questions, the purpose of the research study, the diversity of the sample, the nature of the analysis and the time and resources available. This justification of the sample size, has been reflected in the manuscript as well 9-10

While your methodology mentions the qualitative research method you are using in depth interview and open ended surveys, also useful to specify type of qualitative research method you are using – narrative method or phenomenological or grounded theory etc. Can you explain how cases were assigned and used for the data. The thematic analysis you have used is a good approach to your research. The qualitative design used has now been described in the manuscript as advised. 7

Data analysis procedures should be described in sufficient detail to enable replication.

1. Data analysis has been done selectively – It Should be done for every question from all participants for the questions asked. You have nicely picked put the themes and subthemes, summarize what the participants felt about it. The analysis section has been described to enable replication as suggested 10-11

Discussion and Conclusion refers only to the barrier in regulation and unavailability of transport in the abstract, but you also had the third them on communication which should be mentioned. 

It might be useful to provide examples from other countries how some of the barriers have a solution eg, transport barriers have been taken care of in India through a public private partnership The third theme of barrier in communication has now been reflected in the manuscript as suggested

Examples from other countries are now reflected in the discussion section. 4, 25-26

24-25

Results

The whole section on Participant’s characteristics is missing. Are the participants - those referring or those receiving the referral or both? Please clarify 

The description of participant characteristics is now included 

9

Theme 3 would be better understood with those receiving the referral. 

 The focus of this study was for the referring healthcare facilities; however, this suggestion is accepted for future research. 27

Not clear how table 2 are findings. Seem more like the themes you would have put in your questionnaire to study. Please put the questionnaire as supplementary material. The interview guide has been provided as supplementary material as requested -

Also the columns should be reversed with sub-themes coming after themes. These themes would have come from your literature review and could have started to appear in introduction, but definitely take to material and methods. The title for Table 2 is changed to show that the table reports themes and subthemes. The suggestion to have themes in the first column has been considered accordingly 11-12

In theme 1(ii) – What are the barriers in protocol as understood by the health worker. In theme 2 – what about the availability of staff for the ambulance?

Theme 3: Is discharge form not given for feedback about women’s condition? All comments should be tabulated under a sub-theme to make it easier to follow. 

Also, for each subtheme it is important to know How many responses were received from a given type of facility. Based on just one comment from one provider we cannot make a conclusion that referral protocol has a problem Main is the challenge of shuttling women to an adequate facility, there were no problem with drivers. The table is provided 

This was a qualitative research whereby the interest is the information even if it comes from only one participant and therefore the responses of participants were not counted. From each healthcare facility the plan was to interview 4 but only 2 were interviewed 11-12

-

Further what is the problem and solution perceived by the providers and authors? Eg. for line 203 - could the solution be Grading the condition of the patient by MO and referring based on that. Or line 223-225: off duty emergencies to be allowed by junior staff. Later the MO should be able to assess and sign. Also availability of ambulance at night.

 The referral guideline, require health system levels to be followed. The medical doctor is always the second on call therefore should be available to sign the referral form. This has been reflected in the discussion section. 23

Lines 211-215 need to be more clearly written.

230-233 – Are those the part of referral guidelines.

265-267 – could ambulance be called from higher level facilities. 

There are some good examples of public private partnerships for ambulance services from India. May be useful to discuss in discussion. 

 As suggested the text has been revised for clarity

The solution of transport as per India experience has been reflected in the discussion section as advised 13

24

Discussion

A lot of the beginning discussion could be in introduction. Discussion should follow from results. – Identifying the problems for each theme and providing a solution with support from the literature. 

The discussion section has been revised as suggested 

22-27

Other comments:

1. References need to be completely reformatted. Most references do not follow the PLOS One style. 

PLOS uses the reference style outlined by the International Committee of Medical Journal Editors (ICMJE), also referred to as the “Vancouver” style. Example formats are listed below. Additional examples are in the ICMJE sample references. Need to give first 6 authors et. al. 

Example for published papers: Hou WR, Hou YL, Wu GF, Song Y, Su XL, Sun B, et al. cDNA, genomic sequence cloning and overexpression of ribosomal protein gene L9 (rpL9) of the giant panda (Ailuropodamelanoleuca). Genet Mol Res. 2011;10: 1576-1588.

Putting the references in the text also need to be corrected eg. [5,6] and not [5], [6] …several other places as well . Please check the PLOS one manuscript writing guide. 

Reference 31 not mentioned in the text. Need to be checked. Why is therea .before reference 9 and 10.

2. Several grammatical and spelling errors

Line 53, 87, 227 (lent), 327 (where not were), 

472-476, 481-482 – Grammar not clear. and many more 

References have been reformatted to abide to the PLOS One style 

29-34

Comments from Reviewer 1

I reviewed a manuscript titled " Understanding barriers to implementing referral procedures in the rural and semi-urban district hospitals in Tanzania: Experiences from healthcare providers working in maternity units.

The manuscript is well written Compliments appreciated. -

The authors need to correct a few grammatical errors in the manuscript (e.g line 148, 149) The manuscript has been English edited and grammatical errors corrected 1-28

Methodology: The type of sampling strategy used was not clear. The authors stated the definition of convenience sampling on line 129, but did not clearly indicate if this was the strategy used in this study. Information on participant recruitment using convenient sampling strategy is explained in the manuscript as suggested 9

References: Please remove the period (.) in the beginning of reference 4,5,9 and 10 The references section have been revised to abide to the journal referencing style 29-34

Comments from Reviewer 2: 

The study is relevant in the country context and provides policy recommendations to address high MMR in Tanzania and other countries with similar kind of health system challenges. However, the language of the article could have been more reader friendly. I strongly suggest authors to work with a writing editor to address the grammatical errors, improve the flow and readability of the text. We thank the reviewer for the compliment. Further the manuscript has been English edited and revised to ensure it easier for the reader to understand. 1-28

The authors may consider providing more information regarding “where” mothers are dying in the study area, means whether they are dying at home at the facilities, or in transit. Generally, the data related to place of maternal deaths are available in the MDSR (Maternal Death Surveillance and Response) reports. If Tanzania has MDSR system in place, the data could be analyzed to find out the weakest link in the referral system and findings would complement the qualitative findings of the study making it more statistically relevant. This comment has not been addressed because it is beyond the scope of this study -

Authors have tried to refer and divide their findings as per the standard “Three Delays" model, however, it could have been more explicit in a simple and standard language. Authors may please go through an article from Haiti with similar findings, link is given below:

Barnes-Josiah D, Myntti C, Augustin A. The "three delays" as a framework for examining maternal mortality in Haiti. SocSci Med. 1998 Apr;46(8):981-93. doi: 10.1016/s0277-9536(97)10018-1. PMID: 9579750. We thank the reviewer for this reference, which has been very helpful in describing the 3delays model. -

Comments from Reviewer 3: 

This is an important issue to be raised, timely referrals are important to save lives. This research is based on a very small sample size (only 12 interviews) and the sample is also opportunistic. Would suggest the authors to have a bigger sample to be able to draw conclusions to bring a policy change on referrals. The sample size in qualitative studies is usually guided by information saturation. This information has been included in the manuscript to justify the sample size. 9-12

The authors should do an English language check. The manuscript has been English language checked 1-28

Additional comments from editor-Amy Kiss

If there are no restrictions, please upload the minimal anonymized data set necessary to replicate your study findings to a stable, public repository and provide us with the relevant URLs, DOIs, or accession numbers. Please see http://www.bmj.com/content/340/bmj.c181.long for guidelines on how to de-identify and prepare clinical data for publication. For a list of recommended repositories, please see https://journals.plos.org/plosone/s/recommended-repositories. You also have the option of uploading the data as Supporting Information files, but we would recommend depositing data directly to a data repository if possible. All relevant data are in the paper and its supporting files have shared. 

 -

---

## [Decision Letter · Decision Letter 1]

19 Jul 2021

Understanding barriers to implementing referral procedures in the rural and semi-urban district hospitals in Tanzania: Experiences of healthcare providers working in maternity units

PONE-D-20-36193R1

Dear Dr. Mselle,

We’re pleased to inform you that your manuscript has been judged scientifically suitable for publication and will be formally accepted for publication once it meets all outstanding technical requirements.

Kind regards,

Charu C Garg, Ph.D.

Academic Editor

PLOS ONE

Additional Editor Comments (optional):

Thank you for taking into account all the reviewers comments and revising your paper. However, there are still some editing that is required. There are still some spellers - powerlessness in abstract, line 216- interviewed to be replaced with suitable word. Please check carefully for all grammatical errors. 

Also, while addressing all the comments, some parts of the paper have become very descriptive and would be helpful if some sections can be made more crisp - especially the introduction and discussion. Please see if some unnecessary sentences can be removed

The references still need some work, eg. ref. 22 - missing journal name, ref. 31; reference 1 full names given - use Kassebaum N, Steiner C, Murray CJL, .......; please check all references
---

## [Editor Report · Acceptance letter]

17 Aug 2021

PONE-D-20-36193R1 

Understanding barriers to implementing referral procedures in the rural and semi-urban district hospitals in Tanzania: Experiences of healthcare providers working in maternity units 

Dear Dr. Mselle:

I'm pleased to inform you that your manuscript has been deemed suitable for publication in PLOS ONE. Congratulations! Your manuscript is now with our production department. 

Kind regards, 

on behalf of

Dr. Charu C Garg 

Academic Editor

PLOS ONE